# Strong Tracking PHD Filter Based on Variational Bayesian with Inaccurate Process and Measurement Noise Covariance

**DOI:** 10.3390/s21041126

**Published:** 2021-02-05

**Authors:** Zhentao Hu, Linlin Yang, Yong Jin, Han Wang, Shibo Yang

**Affiliations:** 1School of Artificial Intelligence, Henan University, Kaifeng 475004, China; hzt@henu.edu.cn (Z.H.); jy@henu.edu.cn (Y.J.); 2School of Computer and Information Engineering, Henan University, Kaifeng 475004, China; wanghan_henu@163.com (H.W.); ysb@vip.henu.edu.cn (S.Y.)

**Keywords:** PHD filter, strong tracking, variational bayesian approximation, GIW joint distribution, inaccurate process and measurement noise covariance

## Abstract

Assuming that the measurement and process noise covariances are known, the probability hypothesis density (PHD) filter is effective in real-time multi-target tracking; however, noise covariance is often unknown and time-varying for an actual scene. To solve this problem, a strong tracking PHD filter based on Variational Bayes (VB) approximation is proposed in this paper. The measurement noise covariance is described in the linear system by the inverse Wishart (IW) distribution. Then, the fading factor in the strong tracking principle uses the optimal measurement noise covariance at the previous moment to control the state prediction covariance in real-time. The Gaussian IW (GIW) joint distribution adopts the VB approximation to jointly return the measurement noise covariance and the target state covariance. The simulation results show that, compared with the traditional Gaussian mixture PHD (GM-PHD) and the VB-adaptive PHD, the proposed algorithm has higher tracking accuracy and stronger robustness in a more reasonable calculation time.

## 1. Introduction

Multi-target tracking is one of the core issues in information fusion research, which has been widely used in civil and military fields such as aerospace, electronic information, and control engineering. It mainly estimates the number and state of the target through the data obtained by sensors, where the target is possible to be born, die, and derive at any time. Traditional multi-target tracking methods require data association when tracking multiple targets, such as joint probabilistic data association (JPDA) [1,2,3] and multi-hypothesis tracking (MHT) [4,5], but they can only deal with a fixed number of targets. As the number of targets increases, the calculation amount of these algorithms increases exponentially, which seriously affects the real-time performance.

### 1.1. Related Work

In recent years, the multi-target tracking algorithm based on random finite set (RFS) theory does not require complex data association when tracking multiple targets, so it has attracted widespread attention from scholars [6,7,8]. The probability hypothesis density (PHD) propagates the first-order statistical moment of the RFS, which is developed to alleviate the computational intractability as a recursion [9]. At present, the existing closed-form solutions of PHD filters mainly include particle filter PHD [10,11,12] and Gaussian mixture PHD filter [13,14]. However, these algorithms only have better performance in multi-target tracking systems where the noise variance is known.

Due to the different characteristics of sensors, it is difficult to accurately obtain statistical information of noise in practical applications. For the problem of unknown noise statistics, the variational Bayesian (VB) approximation method is widely used to estimate the state of linear Gaussian systems. Zhang et al. proposed an improved PHD filter. It introduced the VB method into the PHD recursion and derived the closed-form solution of the improved PHD filter for the linear Gaussian multi-target model by using the inverse gamma (IG) and Gaussian mixture distribution [15]. An adaptive cubature Kalman-VB-PHD (ACK-VB-PHD) filter was deduced by Yuan et al., which was based on the PHD filter and used the Cubature Kalman filtering (CK) to approximate the nonlinear measurement model. The measurement noise covariance distribution was described by the inverse Wishart (IW) distribution. Then, it iteratively estimated the measurement noise covariance and multi-target state joint posterior density through the VB approximation technique [16]. Li et al. took the prior gamma distribution for noise parameters in PHD filter, so that the intensity of prediction and update can be represented by a mixture of Gaussian gamma terms. When the target state and noise parameters were coupled in the likelihood function, the VB method was used to derive the approximate distribution [17]. However, using gamma distribution as the prior distribution of measurement noise limits covariance matrix of measurement noise to being a diagonal matrix. Additionally, the above algorithms solved one problem. Only the inaccurate measurement noise covariance was considered.

In real scenes, not only the measurement noise but also the process noise are usually unknown and time-varying. Then, some scholars put forward the use of the VB method to estimate the measurement noise while increasing the estimation of the unknown process noise. Huang et al. proposed a novel variational Bayesian-based adaptive Kalman filter (VBAKF). It selected the IW prior distribution for the prediction error covariance and the measurement noise covariance and used the VB method to derive the target state, the state prediction covariance matrix, and the amount measurement noise covariance matrix [18]. Li et al. proposed a robust Poisson multi-Bernoulli mixture (PMBM) filter to jointly estimate the target motion state, the corresponding state covariance, and the measurement noise covariance. The IW distribution was selected as the conjugate prior distribution of Gaussian. This algorithm introduced VB to ensure the conjugacy to obtain the approximate posterior density of the augmented state [19]. However, due to the simultaneous variation iteration of the measurement noise covariance matrix and predicted error covariance matrix, the above algorithms are time-consuming.

### 1.2. Contributions

Based on the above research, this paper proposes the strong tracking PHD-based on a VB filter in the case of the inaccurate process noise covariance and measurement noise covariance slowly time-varying. This article mainly has the following characteristics:(a)Strong tracking filter is introduced under the framework using VB approximation. Then, the prediction state covariance is modified in real time by the fading factor from strong tracking. The factor not only enhances the function of observation in Kalman filtering but also corrects the influence of inaccuracy of process noise on state covariance.(b)This paper uses IW distribution to model only measurement noise, compared with the literature [18,19] that uses IW distributions as prior distributions of prediction state covariance and measurement noise covariance. It reduces the time consumed by the iterative process and combines the modified predictive state covariance, thus balancing the computation time and tracking accuracy. The kinematic state obeys Gaussian distribution, and then, the augmented state is modeled as the GIW joint distribution.(c)Three scenarios with different parameters are designed to evaluate the performance of filters from the four aspects of Optimal Sub-pattern Assignment (OSPA) error, localisation error, cardinality error, and calculation time. The proposed algorithm is compared to the original GM-PHD filter and the VB adaptive PHD filter with optimal tracking performance and robustness.

### 1.3. Paper Organization and Notation

The main distributions in the remaining chapters of this article are as follows. Section 2 briefly describes the traditional Gaussian mixture PHD filter. Section 3 presentes the proposed filter in this paper. The basic principle of the VB approximation is given in Section 3.1. Section 3.2 introduces the strong tracking algorithm on the VB basis. Then, Section 3.3 shows the derivation details, which are the GIW implementation of the proposed filter. Section 4 shows the design of a multi-target tracking simulation experiment to verify the tracking accuracy and reasonable calculation time of the filter under different parameter scenarios. Section 5 summarizes the contents of this article as a whole.

In this paper, matrices are indicated in bold, such as m. The Kronecker product of matrices is denoted by ⊗. $m^{T}$ is the transposition of matrix m. $∥m∥$ and $tr\left(m\right)$ calculate the two norm and trace of matrix m, respectively. The Kullback–Leibler divergence between q(x) and p(x) is expressed as $\mathrm{KLD}\left(q\left(x\right)\|p\left(x\right)\right)\overset{\Delta }{\;=}\int q\left(x\right)\log\frac{q(x)}{p(x)}dx$.

## 2. Traditional Gaussian Mixture PHD Filter


In the monitoring area, there are M(k) target states and N(k) measurement values at time *k*; the multi-target state set and multi-target observation set are denoted as $X_{k}=\left\{x_{k,1},\cdots ,x_{k,M(k)}\right\}$ and $Z_{k}=\left\{z_{k,1},\cdots ,z_{k,N(k)}\right\}$, respectively. Assuming that the multi-target state set at time step k−1 is $X_{k-1}$, then the current time $X_{k}$ and $Z_{k}$ are expressed as follows:(1)$$X_{k}=\left[\underset{x\in X_{k-1}}{⋃}S_{k|k-1}\left(x\right)\right]⋃\left[\underset{x\in X_{k-1}}{⋃}B_{k|k-1}\left(x\right)\right]⋃\Gamma_{k}$$
(2)$$Z_{k}=\left(\underset{x\in X_{k}}{⋃}\Theta \left(x\right)\right)⋃\kappa_{k}$$
where $S_{k|k-1}\left(x\right)$ is the RFS of multi-target states from $X_{k-1}$ at time k−1 to *k*,$B_{k|k-1}\left(x\right)$ and $\Gamma_{k}$ represent RFS derived and newly born, and $\Theta_{k}\left(x\right)$ and $\kappa_{k}$ are the RFS of observations generated by $X_{k}$ and clutter, respectively.

PHD filter is obtained by the first moment approximation of the posterior multi-target state, but its recursion does not admit closed form solutions [10]. In the linear Gaussian multi-objective system, the implementation of the Gaussian mixture is used to obtain the analytical solution of Bayesian integration and the process can more clearly show how the Gaussian component is analytically propagated to the next moment. Assume that each target follows the linear Gaussian transition model and the linear Gaussian observation model; the posterior PHD at time *k* subject to the Gaussian mixture of the form is as follows:(3)$$v_{k-1}\left(x\right)=\sum_{i=1}^{J_{k-1}}w_{k-1}^{\left(i\right)}N\left(x;m_{k-1}^{\left(i\right)},P_{k-1}^{\left(i\right)}\right)$$
where $J_{k-1}$ is the Gaussian component at time k−1, $w_{k-1}^{\left(i\right)}$ is the weight corresponding to the ith Gaussian component, and its sum is the estimated target number. $N\left(:;m,p\right)$ is the Gaussian density of the mean m and the covariance p. $m_{k-1}^{\left(i\right)}$ and $P_{k-1}^{\left(i\right)}$, respectively, are the target state and state covariance at time k−1. The posterior information of time k−1 is recursively estimated by Kalman filter implementation to obtain the target PHD of time *k*. The predicted intensity $v_{k|k-1}\left(x\right)$ is expressed as follows:(4)$$v_{k|k-1}\left(x\right)=\sum_{i=1}^{J_{k|k-1}}w_{k|k-1}^{\left(i\right)}N\left(x;m_{k|k-1}^{\left(i\right)},P_{k|k-1}^{\left(i\right)}\right)$$

$J_{k|k-1}$ predicts the Gaussian component and its corresponding prediction weight $w_{k|k-1}^{\left(i\right)}=P_{s,k}w_{k|k}^{\left(i\right)}$. $P_{s,k}$ is used to denote target survival probability. The recursive estimation of $m_{k|k-1}^{\left(i\right)}$ and $P_{k|k-1}^{\left(i\right)}$ uses Kalman filter state and state covariance prediction formulas. $v_{k}\left(x\right)$ represents the intensity of multi-target update recursion at time *k*.
(5)$$v_{k}\left(x\right)=\left(1-P_{D,k}\right)v_{k|k-1}\left(x\right)+\sum_{z\in Z_{k}}\sum_{i=1}^{J_{k|k-1}}w_{k}^{\left(i\right)}\left(z\right)N\left(x;m_{k}^{\left(i\right)}\left(z\right),P_{k}^{\left(i\right)}\right)$$
where $P_{D,k}$ is the detection probability. Updated state $m_{k}^{\left(i\right)}$ and state covariance $P_{k}^{\left(i\right)}$ are obtained by prior information on the time *k* and Kalman gain.

The GM-PHD filter usually uses the given process noise covariance $Q_{k}$ and measurement noise covariance $R_{k}$, but it is difficult to reflect the time-varying situation of the real environment under fixed noise statistics. In the process of multi-target tracking, the GM-PHD filter uses an inaccurate $Q_{k}$ to generate an inaccurate predicted state covariance $P_{k|k-1}$. At this time,$P_{k|k-1}$ and inaccurate $R_{k}$ cause the Kalman gain $K_{k}$ to be inaccurate, which eventually affects the Gaussian parameter $m_{k}$ and $P_{k}$, resulting in an inaccurate $X_{k}$.

## 3. Strong Tracking PHD Filter Based on VB Approximation


### 3.1. VB Approximation


When the noise statistics in the actual scene are unknown and slowly time-varying, the probability density of the inaccurate $R_{k}$ and the single target state $X_{k}$ are estimated jointly by the VB approximation. Assuming that they are independent of each other, the joint posterior probability density function (PDF) according to Bayesian rule is as follows:(6)$$p\left(X_{k},R_{k}|Z_{1:k}\right)=\frac{g_{k}\left(Z_{k}|X_{k},R_{k}\right)p_{k|k-1}\left(X_{k},R_{k}|Z_{1:k-1}\right)}{\int g_{k}\left(Z_{k}|X_{k},R_{k}\right)p_{k|k-1}\left(X_{k},R_{k}|Z_{1:k-1}\right)dX_{k}dR_{k}}$$
where $g_{k}\left(Z_{k}|X_{k},R_{k}\right)$ is the likelihood function of $R_{k}$ and $X_{k}$. Due to the unknown posterior PDF of $R_{k}$, it is difficult to obtain an analytical solution. For the convenience of calculation, the VB approximation is used to find the free-form approximate parameter distribution of $p\left(X_{k},R_{k}|Z_{1:k}\right)$ [20], which can be written as follows:(7)$$p\left(X_{k},R_{k}|Z_{1:k}\right)\approx q\left(X_{k}\right)q\left(R_{k}\right)$$
where $q\left(\cdot \right)$ represents the approximate posterior PDF of $p\left(\cdot \right)$. $q\left(X_{k}\right)$ and $q\left(R_{k}\right)$ are obtained by minimizing the KLD between the approximate posterior PDF and the true posterior PDF:(8)$$\left\{q\left(X_{k}\right),q\left(R_{k}\right)\right\}=\arg\min\mathrm{KLD}\left(q\left(X_{k}\right)q\left(R_{k}\right)\left|\right|p\left(X_{k},R_{k}|Z_{1:k}\right)\right)$$

The variational parameters of $q\left(X_{k}\right)$ and $q\left(R_{k}\right)$ are coupled. The method of literature [21] is used to solve the fixed-point iteration, and the iteration converges to the local optimum, which are calculated as follows:(9)$$q\left(x_{k}\right)\approx q^{\left(N\right)}\left(x_{k}\right)=N\left(x_{k};m_{k}^{\left(N\right)},P_{k}^{\left(N\right)}\right)$$
(10)$$q\left(R_{k}\right)\approx q^{\left(N\right)}\left(R_{k}\right)=IW\left(R_{k};u_{k}^{\left(N\right)},U_{k}^{\left(N\right)}\right)$$
where N is the number of variational iterations and $\mathrm{IW}\left(\cdot \right)$ is the IW PDF with the dof parameter $u_{k}^{\left(N\right)}$ and the inverse scale matrix $U_{k}^{\left(N\right)}$. Selecting the IW distribution as the conjugate prior of Gaussian distribution can avoid the restriction that the measurement noise covariance matrix must be a diagonal matrix due to the use of the inverse gamma (IG) distribution.

### 3.2. Strong Tracking Principle With VB Approximation


In order to improve the tracking ability of uncertain system model and to enhance the accuracy of posterior probability density PDF obtained by VB approximation, the fading factor $\eta_{k}$ of strong tracking principle is needed. Its main function is to adjust the state prediction covariance $P_{k|k-1}^{}$ through $\eta_{k}$ to correct the gain $K_{k}^{}$ in real time, to force the residual sequence to be orthogonal, and to resist the performance degradation caused by the uncertain process noise. The revised state prediction covariance is as follows:(11)$$P_{k|k-1}^{*}=\eta_{k}F_{k-1}P_{k-1}F_{k-1}^{T}+Q_{k-1}$$
and $\eta_{k}$ is defined as follows:(12)$$\eta_{k}=\max\left[1,\frac{\mathrm{tr}\left(V_{k}-H_{k-1}Q_{k-1}H_{k-1}^{T}-\beta R_{k}\right)}{\mathrm{tr}\left(H_{k-1}F_{k-1}P_{k-1}F_{k-1}^{T}H_{k-1}^{T}\right)}\right]$$
where β is the weakening factor, which makes the estimation result smoother, and $tr\left(\cdot \right)$ represents the matrix trace. The output residual sequence covariance $V_{k}$ is given by the following:(13)$$V_{k}=\left\{\begin{array}{cc} \gamma_{k}\gamma_{k}^{T} & k=1\\ \frac{\zeta V_{k-1}+\gamma_{k}\gamma_{k}^{T}}{1+\zeta } & k>1 \end{array}\right.$$
where $\gamma_{k}$ is the residual sequence. The forgetting factor ζ improves the influence of the residual sequence and enhances its role in the filter, usually taken in 0.9≤ζ≤1 [22].

In Equation (Equation 12), $R_{k}$ is fixed at each moment in the iterative process of the traditional strong tracking algorithm, but inaccurate $R_{k}$ affects the accuracy of $\eta_{k}$ in the changing environment [23].To solve the above problem, the same form $q\left(R_{k}\right)=\mathrm{IW}\left(R_{k};u_{k},U_{k}\right)$ is introduced after the Bayesian inference, and the measurement noise covariance $R_{k}$ can be expressed as follows:(14)$$R_{k}^{}=\left\{\begin{array}{cc} \frac{U_{0}}{u_{0}-d_{R}-1} & k=1\\ R_{k-1} & k>1 \end{array}\right.$$$d_{R}$ is the dimension of measurement noise covariance. Since $R_{k}$ changes slowly and the range of change is not drastic, the estimated value at the last moment still has great reference value, and $R_{k}$ as the real-time factor of $\eta_{k}$ can correct $P_{k|k-1}^{}$ more accurately. The modified state prediction covariance $P_{k|k-1}^{*}$ not only improves the tracking performance but also reduces the influence of process noise on the estimation results and improves the robustness of uncertain systems.

### 3.3. The GIW Implementation of The VB-Based Strong Tracking PHD Filter


Combining the above theories, this paper introduces the principle of strong tracking on the basis of the VB adaptive PHD filter and then the GIW implementation of the VB-based strong tracking PHD filter is derived for linear multi-target uncertain systems. The prediction state covariance is corrected in real time by the fading factor. The GIW joint distribution is selected to variation approximate the posterior PHD of $x_{k}$ and $R_{k}$ produced by the measurement set. Assuming that the augmented state of a single target is expressed as $\phi_{k}\overset{\Delta }{\;=}\left(x_{k},R_{k}\right)$, the GIW distribution model of the joint probability density is $p_{k}\left(\phi_{k}\right)=N\left(x_{k};m_{k},P_{k}\right)IW\left(R_{k};u_{k},U_{k}\right)$

#### 3.3.1. Prediction

The GIW mixed form of the posterior intensity PHD at time k−1 can be expressed as follows:(15)$$v_{k-1}\left(\phi_{k-1}\right)=\sum_{i=1}^{J_{k-1}}\left[w_{k-1}^{\left(i\right)}N\left(x_{k-1};m_{k-1}^{\left(i\right)},P_{k-1}^{\left(i\right)}\right)IW\left(R_{k-1};u_{k-1}^{\left(i\right)},U_{k-1}^{\left(i\right)}\right)\right]$$

Using a one-step prediction of the posterior intensity PHD at a time by the multi-objective Bayesian, the components of the predicted intensity are the same as the Equation (1).
(16)$$v_{k|k-1}\left(\phi_{k|k-1}\right)=v_{S,k|k-1}\left(\phi_{S,k|k-1}\right)+v_{\beta ,k|k-1}\left(\phi_{\beta ,k|k-1}\right)+\delta_{k}\left(\phi_{k}\right)$$
(17)$$v_{S,k|k-1}\left(\phi_{S,k|k-1}\right)=P_{S,k}\sum_{j=1}^{J_{k-1}}w_{k-1}^{\left(j\right)}N\left(x;m_{S,k|k-1}^{\left(j\right)},P_{S,k|k-1}^{\left(j\right)}\right)IW\left(R;u_{S,k|k-1}^{\left(j\right)},U_{S,k|k-1}^{\left(j\right)}\right)$$
where $P_{S,k}$ is target survival probability, $v_{S,k|k-1}\left(\cdot \right)$ is the surviving target predicted intensity at time *k*, the spawning intensity $v_{\beta ,k|k-1}\left(\cdot \right)$ and the birth intensity $\delta_{k}\left(\cdot \right)$ have the same composition as $v_{S,k|k-1}\left(\cdot \right)$, and the GIW parameters of $v_{S,k|k-1}\left(\phi_{S,k|k-1}\right)$ are derived as follows:(18)$$m_{S,k|k-1}^{\left(j\right)}=F_{k-1}m_{k-1}^{\left(j\right)}$$
(19)$$\gamma_{S,k}^{\left(j\right)}=z_{k}-H_{k}m_{S,k|k-1}^{\left(j\right)}$$
(20)$$u_{S,k|k-1}^{\left(j\right)}=\rho \left(u_{S,k-1}^{\left(j\right)}-d_{R}-1\right)+d_{R}+1$$
(21)$$U_{S,k|k-1}^{\left(j\right)}=\rho U_{S,k-1}^{\left(j\right)}$$
where ρ is a real number, which usually takes the value ρ=1−exp(−4)  [18]. The residual sequence covariance of Equation (Equation 13) is calculated according to $\gamma_{S,k}^{\left(j\right)}$, and then, the improved state prediction covariance $P_{k|k-1}^{*}$ is finally obtained by combining Equations (12) and (14).

#### 3.3.2. Update

The GIW mixed form of the prediction intensity PHD at time *k* can be further written as follows:(22)$$v_{k|k-1}\left(\phi_{k|k-1}\right)=\sum_{i=1}^{J_{k|k-1}}\left[w_{k|k-1}^{\left(i\right)}N\left(x_{k|k-1}^{};m_{k|k-1}^{\left(i\right)},P_{k|k-1}^{\left(i\right)}\right)IW\left(R_{k|k-1};u_{k|k-1}^{\left(i\right)},U_{k|k-1}^{\left(i\right)}\right)\right]$$

Then, the updated intensity PHD forms representatives at the same time:(23)$$v_{k}\left(\phi_{k}\right)=\left(1-p_{D,k}\right)v_{k|k-1}\left(\phi_{k|k-1}\right)+\sum_{z\in Z_{k}}v_{D,k}\left(\phi_{D,k};z\right)$$
where
(24)$$v_{D,k}\left(\phi_{k};z\right)=\sum_{j=1}^{J_{k|k-1}}\left[w_{k}^{\left(j\right)}\left(z\right)N\left(x_{k};m_{k}^{\left(j\right)}\left(z\right),P_{k}^{\left(j\right)}\right)IW\left(R_{k};u_{k}^{\left(j\right)},U_{k}^{\left(j\right)}\left(z\right)\right)\right]$$
(25)$$T_{k}^{\left(j\right)\left(n\right)}=\left(z_{k}-H_{k}m_{k}^{\left(j\right)\left(n\right)}\right){\left(z_{k}-H_{k}m_{k}^{\left(j\right)\left(n\right)}\right)}^{T}+H_{k}P_{k}^{\left(j\right)\left(n\right)}H_{k}^{T}$$
(26)$$u_{k}^{\left(j\right)\left(n+1\right)}=u_{k|k-1}^{\left(j\right)}+1$$
(27)$$U_{k}^{\left(j\right)\left(n+1\right)}=U_{k|k-1}^{\left(j\right)}+T_{k}^{\left(j\right)\left(n\right)}$$
(28)$$R_{k}^{\left(j\right)\left(n+1\right)}={\left\{E^{\left(n+1\right)}\left[R_{k}^{-1}\right]\right\}}^{-1}={\left\{{\left(u_{k}^{\left(j\right)\left(n+1\right)}-d_{R}-1\right)}^{-1}U_{k}^{\left(j\right)\left(n+1\right)}\right\}}^{-1}$$
(29)$$w_{k}^{\left(j\right)\left(n+1\right)}\left(z\right)=\frac{P_{D,k}w_{k|k-1}^{\left(j\right)}N\left(H_{k}m_{k|k-1}^{\left(j\right)},R_{k}^{\left(j\right)\left(n+1\right)}+H_{k}P_{k|k-1}^{*\left(j\right)}H_{k}^{T}\right)}{\kappa_{k}\left(z\right)+P_{D,k}\sum_{\ell =1}^{J_{k|k-1}}w_{k|k-1}^{\left(\ell \right)}N\left(H_{k}m_{k|k-1}^{\left(j\right)},R_{k}^{\left(j\right)\left(n+1\right)}+H_{k}P_{k|k-1}^{*\left(j\right)}H_{k}^{T}\right)}$$
(30)$$K_{k}^{\left(j\right)\left(n+1\right)}=P_{k|k-1}^{*\left(j\right)}H_{k}^{T}{\left(H_{k}P_{k|k-1}^{*\left(j\right)}H_{k}^{T}+R_{k}^{\left(j\right)\left(n+1\right)}\right)}^{-1}$$
(31)$$m_{k}^{\left(j\right)\left(n+1\right)}=m_{k|k-1}^{\left(j\right)}+K_{k}^{\left(j\right)\left(n+1\right)}\gamma_{k}^{\left(j\right)}$$
(32)$$P_{k}^{\left(j\right)\left(n+1\right)}=\left[I-K_{k}^{\left(j\right)\left(n+1\right)}H_{k}\right]P_{k|k-1}^{*\left(j\right)}$$
where $n\in \left\{1,2,...,N\right\}$, *N* is the maximum number of variational iterations. Until $∥m_{k}^{\left(j\right)\left(n+1\right)}-m_{k}^{\left(j\right)\left(n\right)}∥\leq \varepsilon$, the iteration is stopped; otherwise, it continues to loop Equations (26)–(33). $P_{k}^{\left(j\right)\left(N\right)}$,$m_{k}^{\left(j\right)\left(N\right)}$,$u_{k}^{\left(j\right)\left(N\right)}$,$U_{k}^{\left(j\right)\left(N\right)}$,$R_{k}^{\left(j\right)\left(N\right)}$,$w_{k}^{\left(j\right)\left(N\right)}$ are updated and output as the input value of pruning and merging. As time progresses, the GIW mixture component increases. Therefore, after each update, it is necessary to prune the GIW terms for which the existence probability is lower than the threshold *L*. Then, the merging distance of the remaining terms is less than the threshold U, and we can extract the target state finally [24].

#### 3.3.3. GIW-stPHD Algorithm Implementation

In order to more intuitively, introduce the detailed steps of the GIW-stPHD filter. The algorithm flow chart is summarized, as shown in Figure 1 below. The pseudocode of the GIW-stPHD filter is shown in Algorithm 1.
**Algorithm 1:** GIW-stPHD algorithm step flow.1:Give the augmented state $\phi_{k}=\left\{m_{k-1}^{\left(j\right)}\;P_{k-1}^{\left(j\right)}\;u_{k-1}^{\left(j\right)}\;U_{k-1}^{\left(j\right)}\right\}$, $w_{k-1}^{\left(j\right)}$, total step *K*, measurement set $Z_{k}$, number of iterations *N*2:**for **k=1:K**do**3:   Calculating Equations (19) and (20) to get the state prediction value and residual sequence4:   Equations (12)–(14) to obtain the modified covariance of predicted states $P_{k|k-1}^{\left(j\right)*}$5:   Combine Equations (21) and (22) to predict IW distribution parameters6:   Set initial value of variational iteration $m_{k}^{\left(j\right)\left(0\right)}=m_{k|k-1}^{\left(j\right)}$, $P_{k}^{\left(j\right)\left(0\right)}=P_{k|k-1}^{\left(j\right)*}$,$u_{k}^{\left(j\right)\left(0\right)}=u_{k|k-1}^{\left(j\right)}$,$U_{k}^{\left(j\right)\left(0\right)}=U_{k|k-1}^{\left(j\right)}$7:   **for**
n=1:N
**do**8:      Equation (Equation 26) Updates the IW distribution parameters9:      The calculation Equations (29) and (30) obtain $R_{k}^{\left(j\right)\left(n+1\right)}$ and weight $w_{k}^{\left(j\right)\left(n+1\right)}$10:      Calculation Equations (31)–(33) to obtain the kalman gain $K_{k}^{\left(j\right)\left(n+1\right)}$,the state estimate $m_{k}^{\left(j\right)\left(n+1\right)}$ and the State             estimation covariance $P_{k}^{\left(j\right)\left(n+1\right)}$11:      **if**
$∥m_{k}^{\left(j\right)\left(n+1\right)}-m_{k}^{\left(j\right)\left(n\right)}∥\leq \varepsilon$12:         **break**13:      **end if**14:   **end for**15:   **if**
$w_{k}^{\left(j\right)\left(n+1\right)}>L$16:      Choose GIW product terms with a weight greater than *L*17:   **end if**18:   GIW product terms merged in the range of U19:   GIW components with extraction weight greater than the threshold20:   **if**
k<K21:      k=k+1,Returns step 322:   **end if**23:**end for**


## 4. Simulation


### 4.1. Simulation Parameters

A total of 100 time steps were run in the simulation process, and the simulation results were the average after 500 Monte Carlo (MC) trials. For a fair competition, the performance of filters were compared under the same simulation environment of Windows 10-64bit on Intel(R) Core (TM) i5-6500H CPU and 8GB RAM. The dynamic and measurement models were linear Gaussian in this simulation, where the state transition matrix was $F=I_{2}⊗\left(\begin{array}{cc} 1 & T\\ 0 & 1 \end{array}\right)$ and the observation matrix was $H=I_{2}⊗\left(\begin{array}{cc} 1 & 0 \end{array}\right)$. The real time-varying measurement and process noise covariance were as follows:(33)$$Q_{k}=\left[6.5+0.5\cos\left(\pi k/K\right)\right]q\left[\begin{array}{cc} \frac{T^{3}}{3}I_{2} & \frac{T^{2}}{2}I_{2}\\ \frac{T^{2}}{2}I_{2} & TI_{2} \end{array}\right]$$
(34)$$R_{k}=\left[0.1+0.05\cos\left(\pi k/K\right)\right]r\left[\begin{array}{cc} 1 & 0.5\\ 0.5 & 1 \end{array}\right]$$
where *K* is the total simulation time, $q=1\;m^{2}/s^{2}\;$,$r=10\;m^{2}$. *T* is the sampling interval 1 s. The survival probability $P_{S,k}$ is 0.99 for the target from the previous moment to the next moment, and clutter rate is λ=1. The three filters use the same nominal process noise covariance $Q_{0}=I_{4}$ and measurement noise covariance $R_{0}=I_{2}$. $I_{4}$ and $I_{2}$ are four-dimensional and two-dimensional identity matrices, respectively.

The four aspects of OSPA error, localisation error, calculation time, and cardinality error were used as evaluation criteria to compare the performance of the three filters. The OSPA distance [25] has two subsets of *X* and *Z*.The dimensions are *m* and *n*, respectively. When m≤n, it can be defined as follows:(35)$${\bar{d}}_{p}^{\left(c\right)}\left(X,Z\right)={\left(\frac{1}{n}\left(\min_{\pi \in \Pi_{n}}\;\sum_{i=1}^{m}d^{\left(c\right)}{\left(x_{i},z_{\pi (i)}\right)}^{p}+c^{p}\left(n-m\right)\right)\right)}^{1/p}$$Distance sensitivity parameter is 1≤p<∞, and correlation sensitivity parameter is c>0. This article used p=2 and c=100 for simulation. $\Pi_{n}$ refers to all permutations and combinations of $\left\{1,...,n\right\}$. When m>n, ${\bar{d}}_{p}^{\left(c\right)}\left(X,Z\right)={\bar{d}}_{p}^{\left(c\right)}\left(Z,X\right)$. This OSPA error in [25] was decomposed into two components, each of which accounts for localization error $\min_{\pi \in \Pi_{n}}\sum_{i=1}^{m}d^{\left(c\right)}{\left(x_{i},z_{\pi (i)}\right)}^{p}$ and cardinality error $c^{p}\left(n-m\right)$.

### 4.2. Simulation Scenario

In this section, three scenes with different parameters were designed to demonstrate the effectiveness of the proposed GIW-stPHD filter, and the simulations were performed on the 2-D plane. There were four targets in the monitoring range, which appeared at time [1 41 50 71] (s) until time [40 60 70 100] (s), disappearing in sequence, and had uniform motion during survival. In the simulation process, the tracking performance of the GM-PHD filter, the GIW implementation of the VB-based PHD filter (GIW-vbPHD), and the proposed GIW-stPHD were mainly compared. Figure 2 shows the true trajectory of all targets.

#### 4.2.1. Scene Setting of Different Parameters

Table 1 provides different parameters for the three scenes. In scene 1, the detection probability $P_{d,k}$ and tuning parameter τ remained unchanged, and the number of variational iterations *N* kept increasing. Scene 2 and scene 3 were the changes in $P_{d,k}$ and τ, respectively.

#### 4.2.2. Results and Analysis of Different Scenarios

(a)Different number of variational iterations

Figure 3 and Figure 4 show the comparison of the OSPA error and localisation error between the GM-PHD filter, GIW-vbPHD filter, and GIW-stPHD filter when the detection probability and tuning parameter remained unchanged and the variation iteration numbers *N* were 2, 5, and 8, respectively. It can be seen that all three algorithms jump at the target rebirth time and the average error is higher when there are multiple targets at 50 s–60 s than at other times. As a whole, it is obvious that the GIW-stPHD algorithm is superior to the other two algorithms. In the simulation environment with noise is unknown and time-varying, the traditional GM-PHD filter uses constant and inaccurate nominal noise without involving variational iteration. Therefore, its tracking performance in the comparison algorithm is not affected and has large errors when the number of variation iterations changes.

In order to more intuitively describe the tracking performance of the algorithm under different variational iteration times, the average value of the three different algorithms under 500 MC is given in Table 2. The more iteration times the higher tracking accuracy, but the iteration to a certain number of values remain unchanged. Taking the OSPA error as an example, the error of the proposed algorithm is reduced by 21.1% compared with the GIW-vbPHD algorithm when the number of iterations is 2, reduced by 26.7% at N=5. The GIW-stPHD and GIW-vbPHD algorithms are reduced by 7.7 % and 14.3%, respectively, when they are increased from 2 to 5 times. The error remains basically unchanged at N=7 in scene 1, indicating that the tracking accuracy is basically saturated. At this time, even at the expense of computing time, it cannot bring higher tracking accuracy. Therefore, N=5 is a relatively suitable number of variational iterations.

(b)Different detection probabilities

Figure 5 and Figure 6 mainly show the OSPA error and localisation error of the GM-PHD filter, the GIW-vbPHD filter, and the proposed GIW-stPHD filter in scene 2. It can be seen that the three kinds of filtering abruptly change at the same time as in scene 1 and the curves decrease slowly with the decrease in detection probability. When GM-PHD filter is simulated with the minimum detection probability of 0.79, the target is lost. The change in detection probability leads to different measurement values of the target, which ultimately affects the tracking accuracy.

Table 3 gives a more detailed description of the different tracking of the three filters. The localisation errors of the GIW-stPHD filter are reduced by 31.7%, 14.34%, and 3.5% compared with the GIW-vbPHD filter in the detection probability of 0.98, 0.88, and 0.79, respectively. With the decrease of the detection probability, the error of the filter gradually becomes larger. Under different detection probabilities, the tracking accuracy of this proposed algorithm is better than that of the other two algorithms. The greater the detection probability $P_{d,k}=0.98$, the more obvious the advantage of the proposed filter.

(c)Different tuning parameters

Scene 3 mainly focuses on the tracking of GM-PHD filter, GIW-vbPHD filter, and GIW-stPHD filter when the tuning parameters have different values. It can be seen from Figure 7 and Figure 8 that the three different filters all have protrusions at the target rebirth moment. The same filter curves are not very fluctuant under different tuning parameters, but the curve of the GIW-stPHD is lower than that of the other two filters in the same parameter. The change in tuning parameter mainly affects the initial value setting of the dof parameter and the inverse scale matrix in the GIW distribution. In a time-varying noise environment, the GM-PHD filter algorithm does not involve the use of variational inference to approximate, so the change of this parameter has little effect on it. The calculation in Table 4 clarifies that the error of GIW-stPHD filter is the smallest in the scene τ=5, and its OSPA error and localisation error are 28.6% and 34.3% lower than those of GIW-vbPHD filter, respectively.

#### 4.2.3. Performance Analysis of the GIW-stPHD Filter

To sum up, in order to better analyze the overall performance of the proposed GIW-stPHD filter, scene 1 with N=7, scene 2 with $P_{d,k}=0.88$, and scene 3 with τ=5 were selected as case A, case B, and case C, respectively, from the above scenes. In these three different cases, it was compared with the GM-PHD filter and the GIW-vbPHD filter in the two aspects of computational time and cardinality error.

It can be seen from Figure 9 that the cardinality error of the GM-PHD filter, the GIW-vbPHD filter and the proposed GIW-stPHD filter decreases in the three cases. The GM-PHD algorithm takes the shortest time because it does not need to perform variational iteration, but the tracking accuracy is also the worst. Taking case A in Table 5 as an example, the time consumed by GIW-stPHD algorithm is shortened by 0.57s and the error is reduced by 2.5 compared with that of GM-vbPHD algorithm under the same iteration number.

### 4.3. Simulation Complex Scenario

The tracking performances of the GIW-stPHD filter are evaluated under long-term conditions of multiple targets in challenging scenario. Under the same total simulation step, the appearance time and disappearance time of the four targets changed to [1 20 40 60] (s) and [80 80 80 100] (s), respectively. The number of simultaneous targets increases sequentially. Three filters ran 500 MC simulations under different values of the above three parameters. The parameters N=5, $P_{d,k}=0.98$, τ=3 were used as the reference group. Three experimental groups reduced the number of variational iterations and detection probability to N=2 and $P_{d,k}=0.88$, respectively, and changed the tuning parameter to τ=5.

The OSPA errors and localisation errors of the three filters are shown in Figure 10 and Figure 11, and these errors are the averages of the 500 MC experiments. As the number of simultaneous targets increases, the errors corresponding to the three filters are constantly rising, until the error reaches the maximum value at 80 s. When the number of variational iterations decreases to 2, the errors of the proposed filter and the GIW-vbPHD filter increase correspondingly compared with the reference group. With the detection probability of 0.88, the errors of the three comparison algorithms increased significantly. The OSPA error and positioning error of the experimental group for which the tuning parameter changed to 5 are smaller than those of the reference group. The proposed algorithm has minimal errors compared with the comparison filters, which is clearly shown in the figures.

In addition, Table 6 shows that the proposed filter consumes similar time to the GIW-vbPHD filter when the number of variational iterations is 2. However, in other cases, it consumes less time than the GIW-vbPHD filter, with an average of 0.91 s. The GIW mixture component is multiplied when the number of targets increases. This phenomenon increases the workload of pruning and merging steps after updating. Therefore, the computational complexity of the filters increase as the number of targets increases.

### 4.4. Summary of Simulation Results

Through the simulation experiments of the two scenarios under different parameters, the following conclusions are summarized. First, the tracking effect and robustness of the GIW-stPHD filter are optimal when N=5, $P_{d,k}=0.98$ and τ=5 are selected. Combining the above OSPA error and localisation error tables, the fluctuation of detection probability has the greatest impact on filtering performance. The detection probability increased from 0.88 to 0.98, which changed by 11.36%, and the OSPA error changed by 43.08% accordingly. When the tuning parameter changed by 25%, the OSPA decreased by 11.9%. The number of iterations N=2 doubled, thereby reducing the OSPA error by 16.75%. Then, the proposed algorithm achieved an appealing compromise between tracking accuracy and reasonable computing time in the comparison scene in Figure 9 and Table 5. Finally, the simulation results in the complex scenario show that the proposed filter has a stronger adaptability under harsh experimental conditions.

## 5. Conclusions


This paper mainly proposes the GIW-stPHD filter to solve inaccurate process noise and measurement noise in Gaussian linear systems. The algorithm uses the IW distribution as the conjugate prior distribution to model the measurement noise covariance. Then, the fading factor in the strong tracking principle is used to further modify the predicted state covariance to resist the influence of the uncertainty process noise covariance. The VB approach iteratively approximates the posterior probability density. The simulation experiment results show that the proposed GIW-stPHD filter, compared with the GM-PHD filter and the GIW-vbPHD filter, have obvious advantages in the four aspects OSPA error, localisation error, cardinality error, and calculation time under different parameter scenarios. Due to the limitation of the PHD filter itself, it is difficult to resist the strong clutter in the environment. This aspect will be the focus of breakthroughs in the follow-up work.

## Figures and Tables

**Figure 1 sensors-21-01126-f001:**
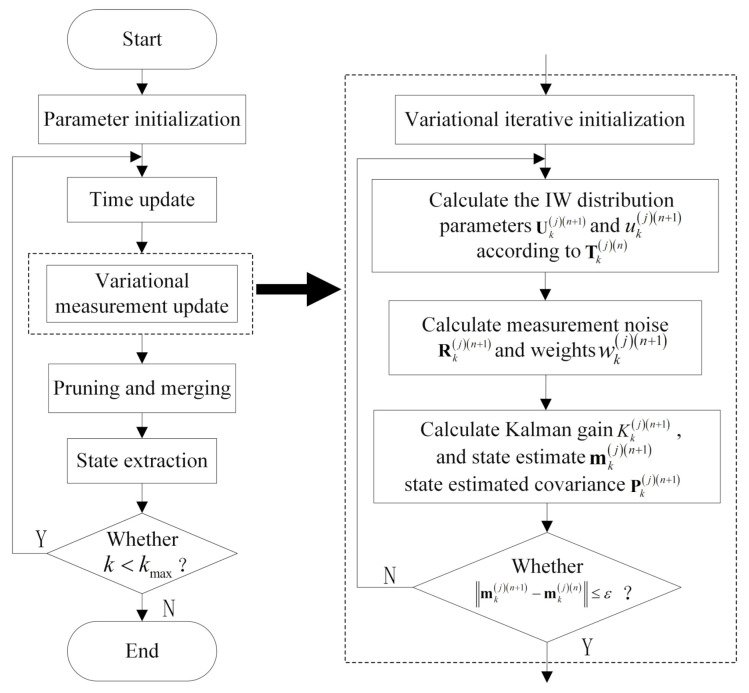
The flow chart of the proposed Gaussian inverse Wishart GIW-stPHD filter.

**Figure 2 sensors-21-01126-f002:**
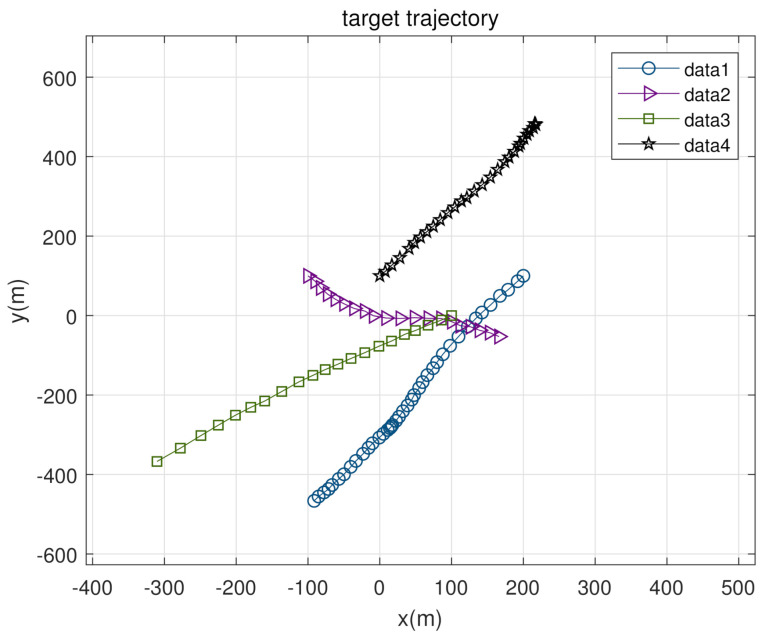
The true trajectory of the target.

**Figure 3 sensors-21-01126-f003:**
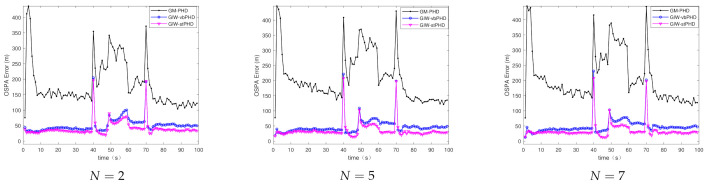
The optimal sub-pattern assignment (OSPA) error in scene 1.

**Figure 4 sensors-21-01126-f004:**
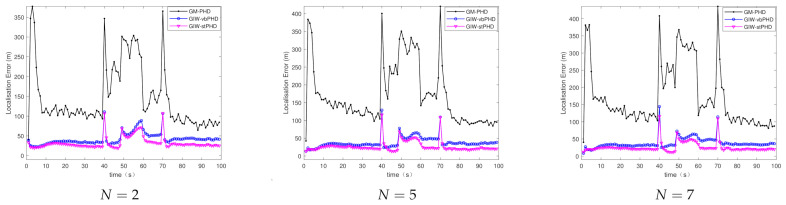
Localisation error in scene 1.

**Figure 5 sensors-21-01126-f005:**
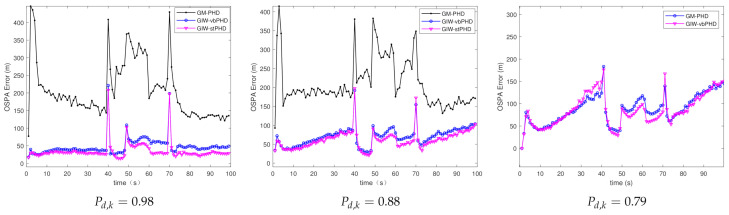
OSPA error in scene 2.

**Figure 6 sensors-21-01126-f006:**
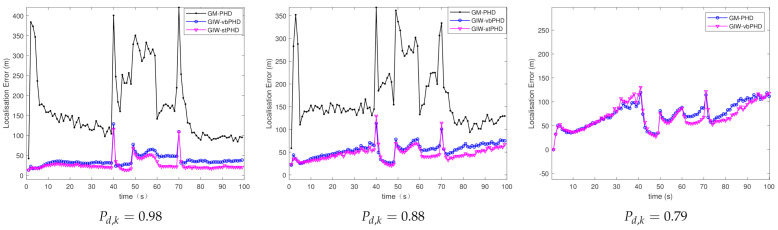
Localisation error in scene 2.

**Figure 7 sensors-21-01126-f007:**
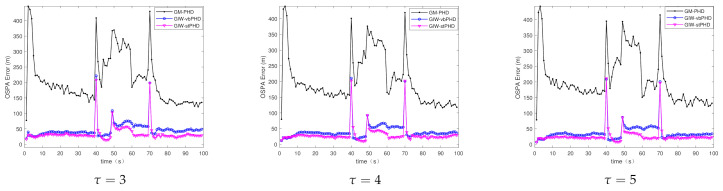
OSPA error in scene 3.

**Figure 8 sensors-21-01126-f008:**
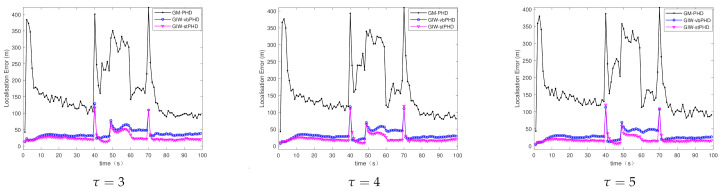
Localisation error in scene 3.

**Figure 9 sensors-21-01126-f009:**
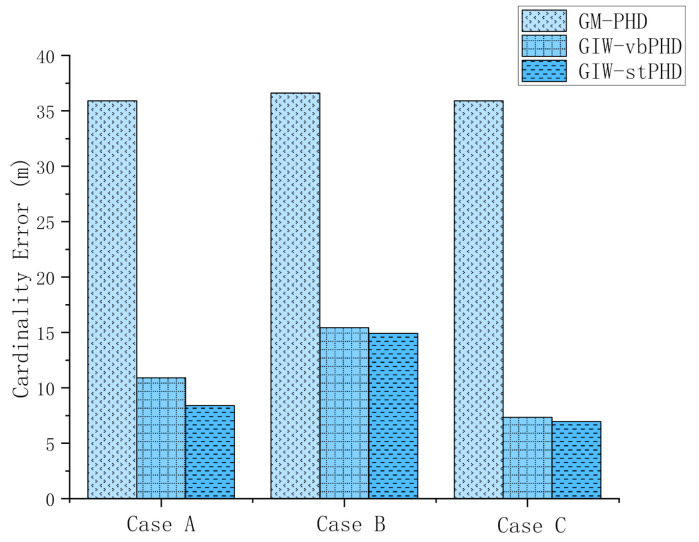
Cardinality error under different filtering.

**Figure 10 sensors-21-01126-f010:**
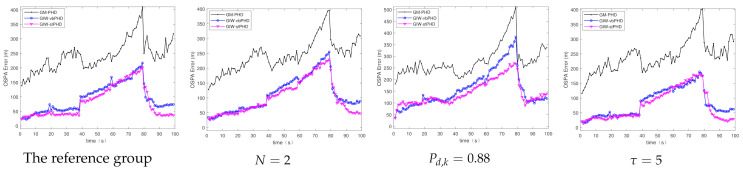
OSPA error.

**Figure 11 sensors-21-01126-f011:**
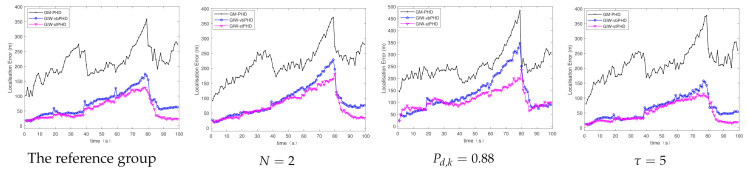
Localisation error.

**Table 1 sensors-21-01126-t001:** Three scenes with different parameters.

Parameter	Scene 1			Scene 2			Scene 3		
$P_{d,k}$	0.98	0.98	0.98	0.98	0.88	0.79	0.98	0.98	0.98
*N*	2	5	7	5	5	5	5	5	5
τ	3	3	3	3	3	3	3	4	5

**Table 2 sensors-21-01126-t002:** OSPA and localisation error under different number of variational iterations.

Filters	OSPA Error $\left(m\right)$			Localisation Error $\left(m\right)$		
	N=2	N=5	N=7	N=2	N=5	N=7
GM-PHD	204.4	206.2	206.3	168.47	145.07	170.48
GIW-vbPHD	51.49	47.5	47.7	42.21	37.46	37.82
GIW-stPHD	40.62	34.79	34.26	32.17	26.62	25.83

**Table 3 sensors-21-01126-t003:** OSPA and localisation error under different detection probabilities.

Filters	OSPA Error $\left(m\right)$			Localisation Error $\left(m\right)$		
	$P_{d,k}=0.98$	$P_{d,k}=0.88$	$P_{d,k}=0.79$	$P_{d,k}=0.98$	$P_{d,k}=0.88$	$P_{d,k}=0.79$
GM-PHD	206.2	208.11	–	145.07	171.41	–
GIW-vbPHD	47.5	69.43	87.59	37.82	54	72
GIW-stPHD	34.79	61.13	86.01	25.83	46.23	69.45

**Table 4 sensors-21-01126-t004:** OSPA and localisation error under different tuning parameters.

Filters	OSPA Error $\left(m\right)$			Localisation Error $\left(m\right)$		
	τ=3	τ=4	τ=5	τ=3	τ=4	τ=5
GM-PHD	206.2	202.6	204.95	145.07	166.73	168.97
GIW-vbPHD	47.5	40.67	37	37.82	32.55	29.67
GIW-stPHD	34.79	29.98	26.4	25.83	22.47	19.49

**Table 5 sensors-21-01126-t005:** Calculating time and cardinality error of three algorithms.

Filters	Calculating Time $\left(s\right)$			Cardinality Error $\left(m\right)$		
	Case A	Case B	Case C	Case A	Case B	Case C
GM-PHD	0.29	0.4	0.29	35.9	36.6	35.9
GIW-vbPHD	1.44	1.52	1.22	10.9	15.42	7.33
GIW-stPHD	0.87	0.93	0.75	8.4	14.9	6.95

**Table 6 sensors-21-01126-t006:** Calculating time.

Filters	Calculating Time $\left(s\right)$			
	The Reference Group	N=2	$P_{d,k}=0.88$	τ=5
GM-PHD	0.21	0.21	0.25	0.23
GIW-vbPHD	1.01	0.53	1.15	1.03
GIW-stPHD	0.88	0.52	0.96	0.89

## Data Availability

No new data were created or analyzed in this study. Data sharing is not applicable to this article.

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
