# Peer review of "Strong Tracking PHD Filter Based on Variational Bayesian with Inaccurate Process and Measurement Noise Covariance"

_sensors, 2021, doi:10.3390/s21041126_

Round 1

Reviewer 1 Report

The  GIW-stPHD  ,presented in this paper, is kind of improvement of PHD. It seems there needs some more analyse to prove merits. And how to evaluate the cost and robustness of this algorithm .

There are several things maybe more explain.  

1) but the noise's covariance is unknown and time-varying in the actual scene.

2)The IW distribution is regarded as the conjugate prior distribution

3)And the GIW joint distribution is selected to variation approximate
the posterior PHD 

4)probability PS,k = 0.99 and clutter λ=1 a,

5) Since the GM-PHD algorithm does not contain variational approximation, the number of iterations does not affect its tracking performance.

Author Response

Dear Ms. Polaris Zhang and dear reviewers,

Thanks very much for taking your time and effort to review this manuscript. Your careful review has helped to make our study clearer and more comprehensive. Here we express our heartfelt thanks! We have studied comments carefully and have made correction which we hope meet with approval.

We would love to thank you for allowing us to resubmit a revised copy of the manuscript and we highly appreciate your time and consideration.

Kind regards,
Ms. Yang

Reviewer 2 Report

Strong Tracking PHD Filter Based on Variational Bayesian with Inaccurate Process and Measurement Noise Covariance

The paper proposes the implementation of a strong tracking PHD filter based on Variational Bayes. The paper is interesting, clear and well written. However, there is a number of issues that must be improved.

¿Are the lines 63 to 76 the contributions of the paper, and the specific differences of the current work with respect to the state of the art? If so, the authors should explicitly say it. Previous works have already used the IW distribution and the VB method and some of the other tools mentioned in these lines. Therefore, it would be necessary to clearly specify which the contributions of the present paper are with respect to recent works in the literature.

How are the main parameters and thresholds of the algorithm tuned? Line 1 of the algorithm consists in this initialization. Please give more details about this step, and perform a sensitivity analysis in the experimental section, so that the reader can check which are the most relevant parameters to the performance of the algorithm and its influence upon it. The influence of some of the parameters is checked now in the experimental section, but as I see it, such experiments should be more exhaustive.

It would be nice to see substantially more challenging tracking experiments, to see the performance of the proposal and to explore to which extent the algorithm keeps on working in a reasonable way when the conditions of the experiment are harsh.

Please include a table of symbols and a table of acronyms at the end of the paper. It would be very useful to make the paper clearer and more readable.

Author Response

(The authors gave the same response as above.)

Reviewer 3 Report

The proposed manuscript presents a strong tracking probability hypothesis density filter based on variational Bayes approximation for multi-target tracking.

The paper is well organized, but has many deficiencies. The main problems are:
-The article is difficult to read and follow due to the weak English.
-Many parts should be described in more detail.
-More complex scenarios should be applied in the simulations.

Detailed comments:
-The manuscript contains many errors in spelling and grammar which need to be corrected. I suggest extensive English proofreading by native speakers.
-Do not use pronoun "we" in the text, use passive voice instead.
-The introduction is well written, but the contributions should be separated from the related work. The differences compared to existing methods should be described in more detail.
-In line 60 the authors state "It better balances tracking accuracy and calculation time.". It should be added (with references) based on comparison with which methods do they make this statement.
-Section 2 is hard to follow. More explanations should be added between the equations.
-Equation 38 is a commonly used formula in related works or it was defined by the authors? If it was defined by the authors, then a detailed explanation should be given, otherwise a citation should be added.
-A description for the Kronecker product used in line 196 should be added to make the paper easier to follow.
-The units are missing in the tables and in the figures.
-How were the errors calculated? Equations and descriptions should be added.
-The method is proposed for tracking in multi-target situations. In the applied scenario there is only a short period when there are 2 targets, in the rest of time there is only 1 target. A more complex scenario should be tested with more targets (2-3-4...). It should be described in detail what effect do multiple targets have on the simulations.
-The applied state and measurement model is too simple, a more complex model could be also tested.
-The calculation times in Table 5 are given in what unit? If they are given in seconds, then how can these algorithms work on real-time systems? The applied frequency is 1Hz (which is pretty low) and the calculation time can be ~0.8-1.5s? It should be also given on what hardware were the calculation times measured.
-The conclusions are missing in Section 4 regarding the parameters. Based on the simulation results what are the optimal parameters?

Author Response

(The authors gave the same response as above.)

Round 2

Reviewer 2 Report

The authors have addressed sufficiently most of the issues that I raised in my previous review. The paper has clearly improved with the revision process.

I have only missed a more complete table of symbols at the end of the paper, detailing the meaning of every symbol that appears throughout the paper. That would make the paper more readable.

Author Response

Dear Ms. Polaris Zhang and dear reviewers,

      Thanks very much for taking your time and effort to review this manuscript. Your careful review has helped to make our study clearer and more comprehensive. Here we express our heartfelt thanks! We have studied comments carefully and have made correction which we hope meet with approval.

   We would love to thank you for allowing us to resubmit a revised copy of the manuscript and we highly appreciate your time and consideration. We wish good health to you, your family, and community.

Kind regards,
Ms. Yang

Reviewer 3 Report

The authors have made the required changes and answered most of the questions, but I still think some parts need to be described in more detail.

Suggestions:
-The authors did not answer my previous question in Concern #9 "It should be described in detail what effect do multiple targets have on the simulations.". I thought that it should be described how does the number targets affect the complexity of the calculations and the computation time (possibly with figures).
-In line 273-274 the authors added the following sentence "Combining the above tables, there is still a conclusion that the proposed filter is more sensitive the change of that parameter." - Which parameter?
-The added complex scenario should be analyzed in more detail, similarly as in section 4.2 (different parameter values, computation time, etc.).
-The conclusions should be removed from section 4.2 (lines 272-278, 289-290) and they should be made based on both simulations (sections 4.2 and 4.3). These conclusions could be added to a new subsection 4.4.
-The authors did not answer my previous questions in Concern #11 "If they are given in seconds, then how can these algorithms work on real-time systems? The applied frequency is 1Hz (which is pretty low) and the calculation time can be ~0.8-1.5s?".

Author Response

(The authors gave the same response as above.)
